# The Effect of Mechanical Vibration during Transport under Model Conditions on the Shelf-Life, Quality and Physico-Chemical Parameters of Four Apple Cultivars

**Dorota Walkowiak-Tomczak** [1], **Natalia Idaszewska** [2], **Grzegorz P. Łysiak** [3,*] and **Krzysztof Bieńczak** [2]

1 Departament of Food Technology of Plant Origin, Poznan University of Life Sciences, ul. Wojska Polskiego 28, 60–637 Poznań, Poland; dorota.walkowiak@up.poznan.pl
2 Institute of Machines and Motor Vehicles, Poznan University of Technology, Piotrowo 3, 60–965 Poznań, Poland; natalia.idaszewska@put.poznan.pl (N.I.); krzysztof.bienczak@put.poznan.pl (K.B.)
3 Department of Ornamental Plants and Pomology, Poznan University of Life Sciences, ul. Dąbrowskiego 159, 60-594 Poznań, Poland
* Correspondence: glysiak@up.poznan.pl; Tel.: +48-61-848-7946

**Abstract:** The study assessed the changes in the quality and physical and chemical parameters of apples of four cultivars ('Gala', 'Idared', 'Topaz', 'Red Prince') subjected to mechanical vibration during transport under model conditions and after storage (shelf-life). Quality changes in apples were evaluated based on skin and flesh colour, total soluble solids, dry matter, firmness, titratable acidity, pH value, total polyphenol content, and antioxidant capacity. The vibration applied at a frequency of 28 Hz caused changes in the above parameters, which were visible also after storage and depended on the cultivar, but often did not show any clear trend or direction or were not statistically significant. The values of the total colour difference factor ΔE showed considerable variations in the skin colour but only small variations in the flesh colour of individual cultivars. Vibration resulted in a decrease in firmness. Variations in dry matter, total soluble solids, pH, and titratable acidity were small, often insignificant. Mechanical vibration and storage led to an increase in the polyphenol content and antioxidant capacity of all studied cultivars. The greatest stability of quality parameters, relatively high content of bioactive compounds, and antioxidant capacity were observed for 'Red Prince'. The lowest quality parameters were noted for 'Gala'. The analysed cultivars continued to show a high level of antioxidant capacity after treatment, which allows the conclusion that they remain a good source of bioactive compounds after transport and short-term storage.

**Keywords:** fruit transport; mechanical damage; physiological disorders; fruit maturity; colour; firmness



## 1. Introduction

Apples are among the most popular fruit species in the world. The worldwide annual production of apples is 86 million tonnes [1]. Because apples are typically grown in a temperate climate and require a winter dormancy period, they are traded internationally and are often transported over long distances. In terms of their share in international trade, they are, after bananas and grapes, the third most frequently transported fruit [2]. The sensitivity of apple flowers to spring frosts, which are typical of the temperate climate zone, results from time to time in lower yields even in countries with high production potential. This also contributes to a high trade volume of this commodity [3]. Apples harvested either too early or too late are easily damaged and susceptible to fungal diseases and physiological disorders [4]. In the case of apples that are too mature at harvest, the increase in vulnerability to mechanical damage is due to the loosening of the flesh structure and the resulting formation of intercellular spaces [5].

Poland is one of the leading producers and exporters of apples and concentrated apple juice in Europe and worldwide [6]. Dessert apples are grown for local consumption and for export, mainly to the EU countries. Within Europe, they are transported mainly by road.

Transport is a key stage in the distribution of fresh food products. At the same time, however, as shown by a number of studies [7–10], transport and transshipment may expose fresh products to serious damage [11]. Fruit is a special case of transport-sensitive cargo. If fruit intended for direct consumption is transported over a distance of tens to several thousand km, it is exposed to mechanical damage and chemical and physiological changes caused by mechanical vibration, the effect of which depends on the time and temperature of transport, the type of packaging, the number of packages (layers) laid out, the type of bodywork and the road quality [12–15]. Mechanical vibration generates continuous impact and friction with neighboring fruit, leading to bruises on its surface. This, in turn, may cause spoilage of the load as a result of rotting fruit. Even minor mechanical injuries may lead to fungal diseases [16]. Studies show that 35% of bruises occur during the harvest and transport of apples [17]. Such damage may result in the rejection of a batch of apples during quality control. The damage level is typically between 10% and 25% [18] but may sometimes result in up to 50% of losses [19]. During simulated transport, up to 80% of apples may be damaged depending on body type, packaging, and position [20]. As consumers demand high-quality fruit, any damage to apples can disqualify this product from further distribution and thus generate substantial financial losses [11,16].

Apples are characterized by valuable nutritional and taste qualities, and, thanks to advanced storage technology, they are available for direct sale and consumption all year round. In addition to their attractive appearance and taste, apples are rich in saccharides, macro- and micronutrients, vitamins, and dietary fibre, including soluble pectins and non-nutrients such as polyphenols. In terms of the content of polyphenols, secondary plant metabolites, apples outperform other common fruit species and contain on average approximately 2000 mg/100 g d.m. of polyphenolic compounds, especially phenolic acids and flavan-3-ols [21,22]. The polyphenol content of the fruit depends on the cultivar and growing conditions [23]. Their level affects the taste and susceptibility to enzymatic browning, especially during transport or processing [24]. Polyphenols as antioxidant compounds affect the health-promoting properties of fruit, including apples. Hence, apples are a recommended diet component as they help prevent many diseases, including cardiovascular, cancer, or hypercholesterolemia [25,26]. Apples have been shown to have a high antioxidant activity in tests with an ABTS cationogen of 24–124 μmol Trolox/100 g d.m., on average about 70 μmol Trolox/100 g d.m. [22,23]. The high nutritional and health-promoting value of apples and their sensory qualities associated with chemical composition, including polyphenol content, may vary during harvest, transport, storage, or processing [27].

This paper describes a study carried out to assess changes in the quality and physicochemical parameters of selected four cultivars of apples as a result of mechanical vibration during transport under model conditions directly after transport and after storage (shelf life).

## 2. Materials and Methods

### 2.1. Plants and Growth Conditions

The study was conducted on four popular apple cultivars: 'Red Prince' (coloured mutant of 'Jonagold'), 'Gala', 'Topaz' and 'Idared'. Fruit was collected in 2019 from a commercial orchard located about 50 km south of Poznan (52°07′28.3″ N 16°58′57.6″ E). The orchard was of full-bearing age.

The harvest occurred on dates determined as the optimum harvest dates (OHD) using the starch test [28] and the sum of active temperatures (growing degree units) method proposed by Łysiak [29].

After harvest, apples intended for storage were graded to eliminate those not meeting the highest commercial quality standard applicable in OECD countries [30]. According to those standards, apples of superior quality ('extra'), have to be intact, sound, clean, and

free of any damage, have a diameter of over 60 mm and a cultivar-specific colour. After harvest, apples were transported to cold storage and stored at 1 °C until all cultivars were collected. The experiment was carried out using 4 boxes of 15 kg of each apple cultivar. The orchard was protected and maintained in line with growing practices recommended for commercial orchards.

### 2.2. Test Stand

The impact of transport on the quality and parameters of apples was assessed under model conditions using a vibration simulator placed in a refrigerated vehicle body with adjustable temperature [31]. A schematic diagram of the vibration simulation stand is shown in Figure 1. The vibration simulator consisted of two basic components: (1) a vibration generating system made up of an inverter-controlled motor, flexible suspension, and vibro-insulation, and (2) an integrated parameter control system. The operation of the vibration simulation stand and the parameters of individual actuators were controlled by monitoring the mechanical quantities (vibration acceleration) in the PC-controlled feedback loop.

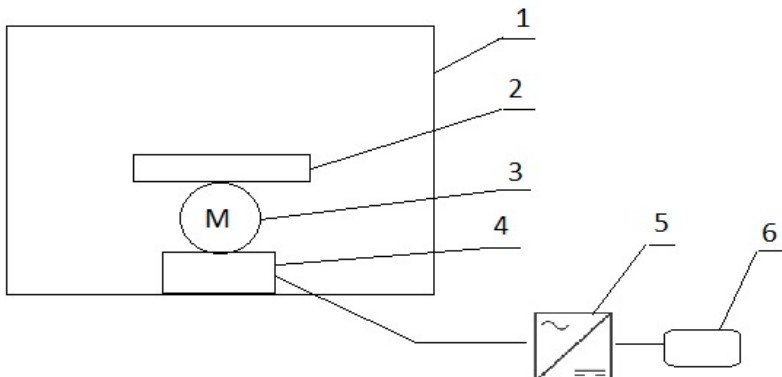

**Figure 1.** Diagram of the vibration simulation stand: 1—stationary vehicle body with adjustable temperature, 2—produce container with fixing elements, 3—0.09 kW motor; 2800 rpm; 230/400 V; 0.58/0.33 A; IMB3, 4—suspension and vibro-insulation, 5—Omron MX2-AB 002-E inverter (SJ200-002NFEF2), 6—computer and software [31].

### 2.3. The Model System of Experiments

Figure 2 shows the experiment workflow. Apples of each cultivar were divided into two batches, one of which was subject to mechanical vibration (transport under model conditions) and the other was a control batch. Apples from both batches were placed in plastic boxes of about 7 kg each. Boxes of the experimental batch were first vibrated at a frequency of 28 Hz for 6 h at a temperature 6 °C (day 0) and then both experimental and control batches were stored in cold storage conditions comparable to those applied in trade and distribution (6 °C) for 14 days to assess the shelf-life of the apples. The vibration frequency was chosen based on the frequencies measured during real-life transport of food by a refrigerated vehicle [32]. The frequency of 28 Hz is within a lower range of the determined predominant frequency values. Apples (both vibrated and control batches) were analysed in terms of their physical and chemical properties on the date of applying vibration and after the 14-day storage period.

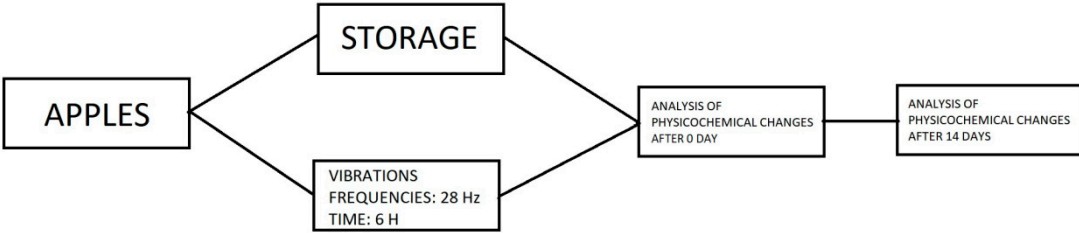

**Figure 2.** Experiment workflow diagram.

### 2.4. Analytical Methods

Three samples of 6 apples each were randomly selected from the experimental batch of each cultivar for the analysis of physical and chemical parameters. The analysis consisted of the measurement of firmness and skin and flesh colour, and after homogenizing the samples (coreless fruit with skin)—of the measurement of pH, titratable acidity, total soluble solids and dry matter. The homogenized samples were also used to prepare methanol extracts (5 g of sample and 50 mL of 80% methanol ($v/v$)) which were analysed for total polyphenol content and antioxidant capacity. After 14 days, the apples were additionally assessed in terms of physiological properties.

#### 2.4.1. Colour Measurement

Colour was measured according to the CIE L*a*b* colour space [33] by a Konica Minolta CR-400 colorimeter (Konica Minolta, Tokyo, Japan) and Color SpectraMagic NX Lite software, in reflected light, using a D65 illuminant and a two-degree observer angle, to determine the L*, a*, b*, C* and h parameters as well as ΔE. The CIE L*a*b* parameters are described (arranged) in a three-dimensional space in which L* denotes the lightness and assumes values from 0 (black) to 100 (white), a* defines the share of the red (+) and green (-) colours, b* defines the share of the yellow (+) and blue (-) colour, C* indicates the chroma and h* stands for the colour hue (angle) and is expressed in degrees (°) to reflect: 0°—red/violet hue, 90°—yellow hue, 180°—blue/green hue, 270°—blue hue, 360°—coincides with 0° [34,35]. The colour of the skin (side without blush) and flesh in the fruit cross-sectional area were measured in 30 repetitions per each sample and the result was expressed as an average value. To compare the colour changes resulting from the shaking (vibration) and storing of fruit, a total colour difference factor ΔE was calculated according to the formula {ΔE = [(ΔL*)2] + [(Δa*)2] + [(Δb*)2]1/2} [35]. The input parameters for ΔE were the $L^0$, $a^0$, $b^0$ values of the control samples on the first day of the experiment.

#### 2.4.2. Firmness Measurement

Firmness was measured using a Fruit Tester 327 EFFEGI FT327 penetrometer (Facchini srl, Alfonsine (Ra), Italy), mounted on stand. The maximum penetration force of a probe of 8 mm in length and 11 mm in diameter, applied to a small area with skin removed, on two opposite sides of the fruit, was recorded. Firmness was measured in 10 repetitions per sample. The results were presented as an average from 30 measurements and expressed in N.

#### 2.4.3. Measurement of Total Soluble Solids

Total soluble solids (TSS) were determined using an ATAGO PAL-1 digital refractometer with automatic temperature compensation (Atago, Tokyo, Japan). The results were shown as an average of nine repetitions per sample and expressed as percentage values.

#### 2.4.4. Measurement of Dry Matter Content

Dry matter content was determined using the gravimetric method in line with ISO 1026-1982 (E) [36]. Samples of about 5 g were dried at 105 °C to obtain solid matter.

The results were presented as an average of nine repetitions per sample and expressed as percentage values.

### 2.4.5. Measurement of pH Value

pH value was measured using a Hanna HI 221 pH meter (Hanna Instruments, Woonsocket, RI, USA) according to the manufacturer's manual. The result was presented as an average of nine repetitions per sample.

### 2.4.6. Measurement of Titratable Acidity

Titratable acidity (TA) was determined using the potentiometric method in line with European Standard EN 12147: 1996 [37]. The results were presented as an average of nine repetitions per sample and expressed as malic acid content (g/L).

### 2.4.7. Measurement of Total Polyphenol Content

Total polyphenol content (TPC) was determined in methanol extracts by means of a spectrophotometric method with Folin-Ciocalteu's reagent [38]. The analysis was carried out on solutions containing 0.7 mL of sample (methanol extract), 0.3 mL of water, 5 mL 0.2 N of Folin-Ciocalteu's reagent, 4 mL of sodium carbonate solution (75 g/L) incubated in the dark for two hours. TPC was measured using a Helios Epsilon spectrophotometer (Thermo Fisher Scientific, Waltham, MA, USA) at a wavelength of 765 nm against blank reagent. The results were presented as an average of nine repetitions per sample and expressed as chlorogenic acid equivalents (mg/100 g d.m.).

### 2.4.8. Measurement of Antioxidant Capacity

Antioxidant capacity of methanol extracts was determined by means of spectrophotometric method using a cationic radical (ABTS+) (2,2′-azinobis-(3-ethylbenzothiazoline-6-sulfonic acid)) [39]. The ABTS+ cation was generated by mixing 7 mM ABTS and 2.45 mM $K_2S_2O_8$ (potassium persulfate or potassium peroxydisulfate) solutions at a ratio of 1:0.5. The ABTS+ cationic radical solution and methanol extract samples were diluted using Phosphate Buffer Solution (PBS) pH 7.4. Absorbance was measured at a wavelength of 734 nm on samples incubated for 6 min at 30 °C against PBS as a reference assay. Antioxidant capacity was determined based on the percentage reduction of absorbance of the ABTS+ cationic radical solution by the sample compared to the reducing power of Trolox (6-hydroxy-2,5,7,8-tetramethylylchromate-2-carboxylic acid). The measurements were conducted using a Helios Alpha spectrophotometer (Thermo Electron Corporation, Waltham, MA, USA) equipped with a water bath for the thermostatting of samples. The results were expressed as an average of nine repetitions per sample and expressed in µmol Trolox/g d.m.

### 2.4.9. Physiological Assessment

Fruit was harvested in line with commercial quality standards and was free from diseases and disorders. After 14 days, the apples were assessed for the presence of physiological disorders, fungal diseases, and bruising. The results were expressed as a percentage share of infected/damaged fruit in the total number of evaluated fruit.

### 2.4.10. Statistical Analysis

The results were analysed by one-way and two-way ANOVA using Statistica 13.3 (StatSoft) software. Significance of differences between average values was determined by Fisher's least significant difference test, at a significance threshold of $p = 0.05$. The correlation between total phenolic content and antioxidant capacity in the samples was calculated using the Pearson correlation coefficient.

## 3. Results and Discussion

### 3.1. Colour

The L*, a*, b*, C* parameters and the hue angle of the evaluated apples varied depending on the cultivar and treatment. The total colour difference factor (ΔE) calculated in relation to control sample 0 within a given cultivar was most often ΔE > 3, which means that the colour differences were noticeable to an average observer. The analysis of the values of individual colour parameters, the ΔE factor, and the significance of statistical differences between average values (results of Fisher's least significant difference test) revealed that they did not follow any clear trend of variation and most probably resulted from the variability of the studied biological material rather than from vibration or storage. The analysed cultivars belong to class B ('Gala', Idared', 'Topaz') and class C ('Red Prince') apples in terms of colour according to the OECD standards [30], which means that at least 50% and 75%, respectively, of the skin area is covered with blush. Therefore, skin colour may vary considerably within a large batch. Such a high percentage of red colouring may mask possible changes in colour (bruising), which would be visible on base colour, i.e., yellow skin. It can be expected that the transport of apples with delicate texture and light creamy green skin, like e.g., 'Papierowka' [26], would cause visible changes in skin colour. Among the studied cultivars, the smallest variations in skin colour parameters (statistically insignificant differences) were observed for 'Idared' apples. However, 'Topaz' and 'Red Prince' apples had darker and redder skin (higher a* values and lower L* and h* values) compared to 'Gala' and 'Idared' apples. When assessing skin colour during the ripening of 'Ligol' and 'Jonagored' apples, Łysiak et al. [40] noted similar L* and C* values, whereas the share of red (a*) was lower and the share of yellow (b*) was higher than those identified in this study, which is connected with the maturity stage. Apples analysed in the current work were fully ripe and were kept in cold storage after harvest till the start of the analyses. Łysiak et al. [40] observed that changes in parameter a* in ripening apples were key for determining the harvest date. The colour parameter values obtained in this study showed the highest colour similarity between 'Idared' and 'Ligol" apples [40] (Table 1).

**Table 1.** Average colour parameters of the skin on the side without blush of control and treated apples.

| Cultivar | Treatment | L* | a* | b* | C* | h | ΔE |
|---|---|---|---|---|---|---|---|
| 'Gala' | Ctr 0 [1] | 68.5 ± 4.13 [ab] | 7.55 ± 6.18 [b] | 33.1 ± 1.83 [b] | 34.5 ± 1.27 [b] | 77.2 ± 10.56 [b] | |
| | Vbr 0 [2] | 69.2 ± 1.62 [ab] | 4.39 ± 2.78 [a] | 41.3 ± 2.22 [c] | 31.4 ± 2.09 [a] | 78.5 ± 4.15 [b] | 8.82 |
| | Ctr 14 [3] | 66.9 ± 4.38 [a] | 7.86 ± 8.11 [b] | 28.0 ± 5.32 [a] | 30.4 ± 3.15 [a] | 73.1 ± 17.49 [a] | 5.37 |
| | Vbr 14 [4] | 69.8 ± 5.46 [ab] | 5.31 ± 6.53 [a] | 29.2 ± 4.11 [a] | 30.1 ± 3.36 [a] | 79.0 ± 13.83 [b] | 4.63 |
| 'Idared' | Ctr 0 | 73.5 ± 3.35 [b] | −5.25 ± 4.17 [a] | 34.9 ± 2.60 [b] | 35.5 ± 2.77 [b] | 98.2 ± 6.88 [b] | |
| | Vbr 0 | 69.4 ± 3.65 [a] | 2.91 ± 5.52 [b] | 30.3 ± 2.28 [a] | 31.0 ± 1.38 [a] | 84.1 ± 10.99 [a] | 10.22 |
| | Ctr 14 | 71.8 ± 5.56 [ab] | 1.01 ± 5.53 [b] | 30.3 ± 2.33 [a] | 30.9 ± 1.28 [a] | 87.7 ± 11.30 [a] | 7.93 |
| | Vbr 14 | 71.6 ± 3.10 [ab] | −2.79 ± 2.49 [a] | 36.9 ± 1.84 [c] | 37.1 ± 1.91 [c] | 94.2 ± 3.83 [b] | 3.69 |
| 'Topaz' | Ctr 0 | 65.4 ± 2.94 [a] | 13.86 ± 5.11 [b] | 35.2 ± 3.47 [a] | 38.3 ± 1.68 [a] | 68.2 ± 8.93 [a] | |
| | Vbr 0 | 65.4 ± 5.66 [a] | 13.22 ± 1.12 [ab] | 36.7 ± 5.24 [ab] | 40.5 ± 1.75 [b] | 69.7 ± 16.16 [ab] | 1.64 |
| | Ctr 14 | 66.5 ± 2.72 [a] | 12.32 ± 4.49 [ab] | 37.2 ± 3.34 [ab] | 39.5 ± 1.95 [ab] | 71.4 ± 7.65 [ab] | 2.76 |
| | Vbr 14 | 66.9 ± 3.12 [a] | 8.98 ± 4.62 [a] | 38.5 ± 4.51 [b] | 40.0 ± 3.10 [b] | 76.4 ± 8.48 [b] | 6.10 |
| 'Red Prince' | Ctr 0 | 62.1 ± 7.47 [a] | 15.22 ± 11.74 [a] | 29.4 ± 6.05 [a] | 35.4 ± 2.85 [b] | 62.9 ± 20.79 [a] | |
| | Vbr 0 | 60.6 ± 2..90 [a] | 12.73 ± 7.49 [a] | 26.1 ± 2.91 [a] | 30.0 ± 2.88 [a] | 64.7 ± 14.36 [a] | 4.34 |
| | Ctr 14 | 62.5 ± 7.77 [a] | 12.82 ± 12.57 [a] | 28.6 ± 5.39 [a] | 34.0 ± 2.64 [b] | 66.4 ± 22.87 [a] | 2.57 |
| | Vbr 14 | 61.6 ± 5.21 [a] | 9.30 ± 5.38 [a] | 28.6 ± 4.22 [a] | 30.6 ± 3.84 [a] | 71.9 ± 10.84 [a] | 5.99 |

Explanation: average values marked with the same letter are not statistically different at *p* = 0.05 within one cultivar (in column) based on a one-way ANOVA. 1—Ctr 0, evaluation after harvest, day 0; 2—Vbr 0, evaluation after vibration, day 0; 3—Ctr 14, evaluation after 14-day storage; 4—Vbr 14, evaluation after vibration and 14-day storage.

Flesh colour parameters of the treated apples of each cultivar differed slightly compared to control apples in all treatments. However, the colour difference ΔE within one cultivar was small and ranged from 0.3 to 2.0, which means that there were no significant differences in flesh colour for an average observer. The analysis of the values of the colour parameters, the ΔE factor, and the significance of differences between average values (results of Fisher's least significant difference test) did not reveal any patterns in the change of apple flesh colour. Although most of those changes were statistically significant, the low ΔE suggests that the flesh colour of treated samples was relative stable and similar to that of control samples within one cultivar. The flesh colour parameter values obtained in this study are comparable to those described in the literature for fresh (non-treated) and sulfite-pretreated apple slices [35]. The presented flesh colour parameter values allow the conclusion that the flesh colour of 'Idared' apples was lighter and closer to the creamy-white colour than that of the other studied cultivars, which was darker and yellower, as can be inferred from the lower L* and h* values and higher a*, b* and C* values obtained for 'Gala', 'Topaz' and 'Red Prince' (Table 2).

**Table 2.** Average colour parameter values of flesh in the cross-sectional area of control and treated apples.

| Cultivar | Treatment | L* | a* | b* | C* | h | ΔE |
|---|---|---|---|---|---|---|---|
| 'Gala' | Ctr 0 | 77.1 ± 1.13 [a] | −2.33 ± 0.55 [b] | 23.1 ± 1.65 [bc] | 23.3 ± 1.67 [bc] | 95.7 ± 1.22 [a] | |
| | Vbr 0 | 77.1 ± 0.47 [a] | −2.82 ± 0.86 [ab] | 22.2 ± 1.17 [a] | 22.4 ± 1.09 [a] | 97.3 ± 2.49 [bc] | 1.11 |
| | Ctr 14 | 78.0 ± 0.87 [b] | −3.01 ± 0.86 [a] | 22.3 ± 0.81 [ab] | 22.5 ± 0.84 [ab] | 97.6 ± 2.12 [c] | 1.42 |
| | Vbr 14 | 76.9 ± 2.05 [a] | −2.51 ± 1.16 [ab] | 23.8 ± 1.45 [c] | 23.9 ± 1.40 [c] | 96.1 ± 2.81 [ab] | 0.73 |
| 'Idared' | Ctr 0 | 79.3 ± 1.04 [a] | −3.07 ± 0.25 [ab] | 15.7 ± 0.72 [b] | 16.1 ± 0.75 [b] | 101.2 ± 0.51 [b] | |
| | Vbr 0 | 79.5 ± 1.64 [a] | −2.79 ± 0.27 [c] | 14.2 ± 1.49 [a] | 14.4 ± 1.50 [a] | 101.2 ± 0.71 [b] | 1.62 |
| | Ctr 14 | 79.0 ± 0.65 [a] | −2.95 ± 0.50 [bc] | 15.8 ± 1.39 [b] | 16.0 ± 1.45 [b] | 100.5 ± 1.01 [a] | 0.30 |
| | Vbr 14 | 79.2 ± 0.55 [a] | −3.21 ± 0.21 [a] | 16.6 ± 0.92 [c] | 16.9 ± 0.94 [c] | 101.0 ± 0.35 [ab] | 0.83 |
| 'Topaz' | Ctr 0 | 74.9 ± 1.62 [ab] | −0.06 ± 0.58 [b] | 25.5 ± 1.44 [a] | 25.5 ± 1.45 [a] | 88.7 ± 1.26 [a] | |
| | Vbr 0 | 74.5 ± 1.47 [a] | −1.12 ± 0.35 [a] | 24.9 ± 1.77 [a] | 24.9 ± 1.77 [a] | 92.6 ± 0.87 [c] | 1.27 |
| | Ctr 14 | 75.4 ± 0.87 [b] | −0.09 ± 0.63 [b] | 25.2 ± 1.40 [a] | 25.2 ± 1.40 [a] | 90.2 ± 1.43 [b] | 0.57 |
| | Vbr 14 | 75.5 ± 0.64 [b] | −0.12 ± 0.30 [b] | 25.4 ± 1.13 [a] | 25.4 ± 1.13 [a] | 90.3 ± 0.68 [b] | 0.60 |
| 'Red Prince' | Ctr 0 | 75.0 ± 1.19 [a] | −1.89 ± 0.51 [b] | 24.1 ± 1.77 [a] | 24.2 ± 1.74 [a] | 94.6 ± 1.41 [a] | |
| | Vbr 0 | 75.8 ± 0.68 [b] | −1.75 ± 0.54 [b] | 24.4 ± 1.54 [a] | 24.4 ± 1.57 [a] | 94.1 ± 1.06 [a] | 0.84 |
| | Ctr 14 | 75.7 ± 0.68 [b] | −2.43 ± 0.46 [a] | 23.6 ± 1.47 [a] | 23.7 ± 1.44 [a] | 95.9 ± 1.38 [b] | 1.03 |
| | Vbr 14 | 76.1 ± 1.09 [b] | −2.11 ± 0.79 [a] | 25.8 ± 2.02 [b] | 26.0 ± 2.04 [b] | 96.9 ± 2.27 [b] | 2.02 |

Explanation: average values marked with the same letter are not statistically different at $p = 0.05$ within one cultivar (in column) based on a one-way ANOVA. 1—Ctr 0, evaluation after harvest, day 0; 2—Vbr 0, evaluation after vibration, day 0; 3—Ctr 14, evaluation after 14-day storage; 4—Vbr 14, evaluation after vibration and 14-day storage.

### 3.2. Firmness

The applied vibration and storage reduced firmness, but the changes were not always statistically significant. Initially, the highest firmness was observed for 'Gala' apples, but this cultivar also showed the biggest drop in firmness after vibration (by 9%), after storage (by 10%) and after vibration and storage (by 17%) relative to the control sample. Considerable changes in firmness were noted for 'Idared' apples—firmness decreased by up to 13% in the vibrated and stored sample. 'Red Prince' apples had the most stable firmness, which dropped by only 1–7%, depending on the treatment, but the changes were not significant (Table 3).

**Table 3.** Average firmness values of control apples, vibrated apples, and apples after 14 days of storage.

| Cultivar | Firmness [N] | | | |
|---|---|---|---|---|
| | **Control 0** | **Vibration 0** | **Control 14** | **Vibration 14** |
| 'Gala' | 58.6 ± 2.90 [b] | 53.0 ± 2.70 [a] | 52.3 ± 3.72 [a] | 48.4 ± 5.34 [a] |
| 'Idared' | 45.3 ± 5.06 [b] | 41.5 ± 2.24 [a,b] | 41.7 ± 2,41 [a,b] | 39.5 ± 2,81 [a] |
| 'Topaz' | 52.8 ± 1.80 [b] | 50.1 ± 2.50 [a,b] | 49.6 ± 3.64 [a,b] | 47.6 ± 2.20 [a] |
| 'Red Prince' | 53.2 ± 1.63 [a] | 52.9 ± 1.72 [a] | 51.4 ± 1.97 [a] | 49.5 ± 6.10 [a] |

Explanation: average values marked with the same letter are not statistically different at $p = 0.05$ within one cultivar (in rows) based on a one-way ANOVA.

The range of firmness values obtained for the four cultivars analysed in this study is lower than that mentioned in the literature for 'Jonagold' apples (65–70 N) [23]. In both studies, firmness decreased during storage. The above-mentioned study reported a decline in the firmness of 'Jonagold' apples by 13–15% after 6 months of ULO storage at 0.5 °C, whereas in this study (standard cold storage at 6 °C) firmness decreased by 3–10% in the control and by 5–9% in the vibrated samples. The decrease in firmness during storage is a natural process commonly described by scholars [41–43]. It is related to the disintegration of complex hydrocolloid compounds, which are responsible for the plant tissue texture, through, among other things, enzymatic degradation of pectin compounds leading to cell wall disintegration and atrophy in the middle lamella [44]. No significant impact of mechanical vibration on the firmness of the treated samples compared to the control was observed, except 'Gala' apples. Probably, the 14-day period set for the assessment of shelf life was too short for any significant firmness differences between control and vibrated samples to occur. From among the tested samples, 'Gala' apples proved to be the most vulnerable to changes in firmness caused by storage and vibration. Other authors also confirm that vibration reduces the firmness of apples [45] (Table 3).

*3.3. Total Soluble Solids, Dry Matter, Titratable Acidity and pH Value*

The content of total soluble solids depended on cultivar and treatment (Table 4). The lowest level of total soluble solids was noted in 'Idared'. Apples of this cultivar are characterized by a relatively low TSS content, which has also been shown in other studies [46]. The highest TSS level was measured in 'Topaz' apples from the control sample and from the sample subjected to vibration and storage. Vibration caused a TSS increase in 'Gala' apples and a TSS decrease in 'Topaz' apples, both from samples not subject to storage. In turn, among the samples kept in cold storage for 14 days only 'Gala' apples showed a decrease in TSS. No statistically significant changes in TSS were observed in the remaining cultivars. The TSS values were consistent with the literature on the subject [22]. The level of total soluble solids is a good indicator of sugar concentration in the fruit, which is strongly affected by storage [46]. Hydrolysis of polysaccharides into monosaccharides or disintegration of cell wall components usually result in an increase in TSS during the storage of apples. In this study, vibration mostly had no influence on the TSS content in apples [47]. Nevertheless, this parameter may drop [48] or rise [44] as a result of vibration or storage. Changes in TSS depend on the duration and conditions of transport and storage and on the cultivar's genetic properties. There were slight, insignificant variations in TSS in the analysed cultivars, including 'Idared' and 'Red Prince'. Probably, metabolic processes and carbohydrate hydrolysis occurred only to a small extent during the relatively short period of storage. 'Idared' and 'Red Prince' apples are characterized by good storability. Changes in TSS highly correspond to changes in firmness because the increase in the content of monosaccharides is related to the disintegration of complex carbohydrates, which are responsible for tissue texture [44]. This is confirmed by the results obtained for

'Red Prince', for which the smallest, insignificant changes in firmness and TSS content were observed (Table 4).

**Table 4.** Total soluble solids (TSS), dry matter content, titratable acidity (TA), and pH value of control and treated apples.

|  | Storage Time | Sample | Cultivar | | | |
|---|---|---|---|---|---|---|
|  |  |  | Gala | Idared | Topaz | Red Prince |
| TSS [%] | 0 | control | 13.41 ± 0.57 [a] | 12.81 ± 0.84 [a] | 15.89 ± 0.70 [b] | 14.32 ± 0.42 [a] |
|  |  | vibration | 14.40 ± 0.69 [bc] | 13.32 ± 0.44 [a] | 14.49 ± 1.02 [a] | 14.40 ± 0.36 [a] |
|  | 14 | control | 14.72 ± 0.35 [c] | 13.11 ± 0.37 [a] | 15.53 ± 0.52 [ab] | 14.14 ± 0.60 [a] |
|  |  | vibration | 13.81 ± 0.52 [ab] | 13.29 ± 0.48 [a] | 14.88 ± 0.47 [a] | 14.49 ± 0.22 [a] |
| Dry matter [%] | 0 | control | 13.85 ± 0.74 [a] | 13.65 ± 1.10 [a] | 15.96 ± 0.46 [a] | 14.93 ± 0.91 [a] |
|  |  | vibration | 14.41 ± 0.66 [a] | 14.15 ± 0.73 [a] | 15.22 ± 0.66 [a] | 15.04 ± 0.55 [a] |
|  | 14 | control | 15.48 ± 0.51 [a] | 14.10 ± 0.42 [a] | 16.37 ± 0.46 [a] | 14.13 ± 1.41 [a] |
|  |  | vibration | 15.37 ± 1.19 [a] | 14.49 ± 0.78 [a] | 15.51 ± 0.84 [a] | 15.05 ± 0.20 [a] |
| pH value | 0 | control | 4.10 ± 0.11 [b] | 3.78 ± 0.14 [c] | 3.54 ± 0.12 [b] | 3.80 ± 0.05 [bc] |
|  |  | vibration | 3.92 ± 0.05 [a] | 3.39 ± 0.05 [a] | 3.40 ± 0.06 [b] | 3.94 ± 0.08 [c] |
|  | 14 | control | 3.87 ± 0.06 [a] | 3.59 ± 0.06 [b] | 3.04 ± 0.07 [a] | 3.79 ± 0.11 [b] |
|  |  | vibration | 4.06 ± 0.06 [b] | 3.72 ± 0.05 [bc] | 3.46 ± 0.09 [b] | 3.59 ± 0.06 [a] |
| TA [g/L] | 0 | control | 2.59 ± 0.15 [ab] | 3.13 ± 0.20 [a] | 6.34 ± 0.08 [ab] | 3.17 ± 0.28 [a] |
|  |  | vibration | 2.72 ± 0.08 [b] | 4.33 ± 0.08 [c] | 6.61 ± 0.08 [b] | 3.08 ± 0.13 [a] |
|  | 14 | control | 2.77 ± 0.15 [b] | 3.71 ± 0.08 [b] | 6.81 ± 0.20 [b] | 3.39 ± 0.08 [a] |
|  |  | vibration | 2.37 ± 0.08 [a] | 3.84 ± 0.08 [b] | 6.21 ± 0.08 [a] | 4.24 ± 0.08 [b] |

Explanation: average values marked with the same letter are not statistically different at $p = 0.05$ within one cultivar (in column) based on a one-way ANOVA.

Dry matter in the studied samples varied depending on cultivar and treatment (Table 4). Just like with TSS, the lowest level of dry matter was found in 'Idared' apples, whereas the highest level was observed in 'Topaz' apples. The dry matter values matched those reported in the literature [22,49]. Neither vibration nor storage caused any significant changes in the dry matter content, irrespective of the cultivar, which was probably due to the relatively short storage period or low physiological activity [44].

The pH value varied depending on cultivar and treatment (Table 4). Compared with control samples, there was a decrease in the pH of the vibrated or stored apples and an increase in the pH of the vibrated and stored apples. The exception was 'Red Prince', where the changes in the pH value showed a reverse direction. The increase in pH in apples (by 4–14% depending on the cultivar) after vibration followed by 14-day storage can be explained by the fact that vibration accelerates metabolic processes in the fruit by increasing the respiration rate [50].

The measured pH values were in agreement with the literature [22] (Table 4). The pH value of the fruit depends mainly on the content of organic acids. Their breakdown during respiration results in lower acidity and pH increase in during storage. However, in the study by Jan et al., pH changed little and insignificantly during the first 30 days of storage [44]. In this study, the same was observed for the vibrated sample of 'Red Prince'.

Titratable acidity assumed the lowest values in the 'Gala' samples and the highest in the 'Topaz' samples, which corresponds to the pH values (the highest for 'Gala' and lower for 'Topaz'). In the above cultivars, TA was lower in the samples subject to vibration than in the control samples before storage and in the vibrated samples after storage. It decreased

(by 6–13%) in the samples subjected to vibration and storage, which went hand in hand with the rise in their pH values. The TA decline is associated with the respiration rate [44]. However, it should be stressed that changes in TA varied in individual cultivars depending on treatment and did not follow the above rule in each cultivar. The increase and decrease in TA observed during storage of fruit in this study was also described by other scholars [51]. Napolitano et al. noted both an increase and decrease in TA in four apple cutivars after four months of cold storage at 2 °C [52]. Other studies did not find any statistically significant changes in TA of four apple cultivars, including 'Gala', during 30-day cold storage at 5 ± 1 °C [44].

The analysis of the above-described fruit parameters after harvest, during transport and storage should also take into account the indirect influence of ethylene, which is synthesized in the fruit during respiration and ripening, especially in stress conditions such as mechanical vibration induced by a moving vehicle. Ethylene initiates various physiological and biochemical processes, mainly respiration, and thus indirectly affects transpiration and disintegration of spare substances. It may influence those processes still on the tree, but this influence is most pronounced during harvest and storage. In climacteric fruit, such as apples, ethylene production increases considerably, even several dozen times, a few days preceding the optimum harvest date and a few days after harvest [53]. To reduce ethylene production, ripening processes, and fruit decay after harvest, the ethylene perception inhibitor 1-methylcyclopropene (1-MCP) is applied at the beginning of storage [54].

### 3.4. Polyphenol Content and Antioxidant Capacity

TPC varied widely, depending on cultivar and treatment (Table 5). The lowest TPC was noted in 'Gala', whereas the highest was found in 'Topaz' (this refers to the average value from all treatments and the results of individual treatments). Apples that were subject to vibration had a higher TPC than control apples. The same was found after 14-day storage. In most cases, those differences were statistically significant. A two-way ANOVA showed that TPC significantly depended on the storage time, and in the case of 'Topaz'—on vibration.

**Table 5.** Total polyphenol content (TPC) in control and treated apples depending on cultivar and storage time.

| | Storage Time | Sample | Cultivar | | | |
|---|---|---|---|---|---|---|
| | | | Gala | Idared | Topaz | Red Prince |
| TPC [mg/100 g d.m] | 0 | control | 479 ± 56.0 [a] | 517.1 ± 33.4 [a] | 763.8 ± 39.3 [a] | 596.6 ± 67.8 [a] |
| | | vibration | 486 ± 9.7 [ab] | 549.8 ± 21.9 [b] | 824.7 ± 85.0 [b] | 620.7 ± 56.4 [ab] |
| | 14 | control | 517 ± 11.7 [bc] | 563.1 ± 36.6 [b] | 802.5 ± 30.2 [ab] | 664.9 ± 21.5 [bc] |
| | | vibration | 540 ± 36.8 [c] | 564.2 ± 20.8 [b] | 840.3 ± 23.2 [b] | 683.5 ± 20.1 [c] |

Explanation: average values marked with the same letter are not statistically different at $p = 0.05$ within one cultivar (in column) and one parameter based on a two-way ANOVA.

Antioxidant capacity varied depending on whether apples were vibrated and stored (Table 6) and varied strongly depending on the cultivar. Similarly as in the case of TPC, the lowest antioxidant capacity was observed in the 'Gala' control sample on day 0, and the highest in the 'Topaz' sample after vibration and storage. Vibrated samples showed higher antioxidant capacity than control samples—both on day 0 and after 14 days of storage. All samples had higher antioxidant capacity after than before storage. Differences between the average values were significant in most of the samples. A two-way ANOVA showed that the changes in antioxidant capacity were significantly affected by both vibration and storage, except 'Topaz' apples, in which case this parameter significantly depended on vibration only.

**Table 6.** Antioxidant capacity of control and treated apples depending on cultivar and storage time.

| | Storage Time | Sample | Cultivar | | | |
|---|---|---|---|---|---|---|
| | | | Gala | Idared | Topaz | Red Prince |
| Antioxidant capacity [μmol Trolox/g d.m.] | 0 | control | $32.6 \pm 3.8$ [a] | $35.2 \pm 2.3$ [a] | $51.9 \pm 2.7$ [a] | $40.6 \pm 4.6$ [a] |
| | | vibration | $33.4 \pm 0.7$ [a] | $37.7 \pm 1.5$ [bc] | $56.6 \pm 5.8$ [b] | $42.6 \pm 3.8$ [ab] |
| | 14 | control | $34.2 \pm 0.8$ [a] | $37.2 \pm 2.4$ [b] | $52.9 \pm 2.0$ [a] | $43.9 \pm 1.4$ [b] |
| | | vibration | $37.7 \pm 2.6$ [b] | $39.4 \pm 1.45$ [c] | $58.7 \pm 1.6$ [b] | $47.7 \pm 1.4$ [c] |

Explanation: average values marked with the same letter are not statistically different at $p = 0.05$ within one cultivar (in column) and one parameter based on a two-way ANOVA.

Mechanical vibration generated during transport makes fresh fruit more likely to undergo physical, chemical, and biological changes during trade and storage. Transport shocks and vibration result directly in visual damage such as abrasion and bruising, which facilitates microbial invasion and multiplication, thus accelerating fruit decay [27,55]. Biochemical changes induced by stress conditions during transport may also include an increase in the content of polyphenols, which are secondary metabolites playing defensive functions in plants, e.g., preventing microbial access and development [56]. The content of polyphenolic compounds in apples also depends on the agricultural and climatic conditions during growth as well as on storage conditions [57].

The TPC obtained in this work is comparable to the data provided by previous studies: 221–476 mg/100 g d.m. in 'Jonagold' [23], 564–748 mg/100 g d.m. in 'Golden Delicious' [58] and 790–1330 mg/100 g d.m. in 'Papierowka' and 'Gold Milenium' [26]. The literature reports both an increase and decrease in TPC in fruit after harvest. During long-term storage in a controlled atmosphere of 'Golden Delicious' apples, polyphenol content was found to have increased in between the 1st and 3rd month and to have decreased afterwards [58]. Polyphenol content was observed to have increased in 'Auksis' apples cold-stored in normal and controlled atmosphere (ultra-low oxygen), including after applying 1-MCP [57]. The same was noted during the storage of e.g., 'Golden Delicious', 'Pinova' and 'Topaz' apples and the proposed explanation was that the increase may have been due to, among other things, the activity of ethylene, which stimulates L-phenylalanine ammonia lyase (PAL), one of key enzymes responsible for the synthesis of polyphenols [56]. Cold storage triggers in apples a synthesis of ACC-oxidase (1-aminocyclopropane-1-carboxylate *oxidase*), an enzyme crucial to the synthesis of ethylene. Ethylene, in turn, induces the synthesis of PAL enzyme, which regulates the biosynthesis of flavonoids [59]. An increase in the polyphenol content may also result from the depolymerization of phenolic compounds to more water-soluble free phenols [52]. During ripening and storage, polyphenols bound to the cell wall evolve into a free form, which can be more efficiently extracted for analytical purposes [59].

Antioxidant capacity of plant materials is significantly correlated with the content of bioactive compounds with antioxidant properties, such as polyphenols or vitamins [56,60–62]. This is also confirmed by this study, in which the correlation co-efficient between TPC and antioxidant capacity was between 0.94 and 0.97 depending on the cultivar, thus demonstrating a high positive correlation. The antioxidant capacity values observed in this study are comparable with those presented in the literature [22,23]. Studies also show that an increase in polyphenol content during storage is accompanied by an increase in antioxidant capacity [23,56,63]. At the same time, it has been noted that keeping apples in room temperature, i.e., in conditions similar to consumers' homes, leads to a drop in their polyphenol content and thus also antioxidant capacity [56]. Generally, there has been a lot of controversy over the relation between the content of phenolic compounds and the antioxidant capacity of food products. Some studies identified no correlation at all, whereas other studies found a strong correlation between antioxidant capacity and the content of phenolic compounds in apple extracts. This discrepancy is probably due to

the differences in the methods used to measure antioxidant capacity as well as different extraction procedures [52].

Biochemical changes occur not only when the fruit ripens on the plant but also after harvest, when such changes are triggered by physiological processes which depend on external conditions. During the transport and storage of apples, as a result of metabolic processes, respiration and transpiration, changes occur in the content and profile of bioactive compounds, including polyphenols. Those compounds are crucial to the plant's resistance to pathogens, so their concentration rises in response to mechanical damage and microbial infections in fruit after harvest [56]. Harvest and transport trigger a physiological reaction of fruit to stress conditions, which is expressed by, among other things, an increase in polyphenol content and antioxidant activity during distribution and storage. The high polyphenol content and antioxidant capacity of the studied samples allow the conclusion that apples remain a valuable and healthy food component even after being transported and kept in cold storage or on a shop shelf.

### 3.5. Changes Caused by Diseases and Disorders

Postharvest diseases of pome fruit result in substantial economic losses during storage worldwide every year [64]. In the study, the occurrence of fungal diseases, physiological disorders, and visible physical damage of fruit flesh depended on the cultivar and treatment (Figure 3). The apples were thoroughly selected for the experiment and of top quality, therefore virtually all apples were healthy during a preliminary assessment, and any diseases, disorders, or damage were identified only in isolated cases (bitter pit in 'Red Prince' and bull's eye rot in 'Idared') and were most probably due to grading errors. Apples vibrated and assessed on the same day did not have any fungal diseases or physiological disorders; only bruising was identified, probably as an effect of treatment. After 14-day storage, both vibrated and control apples had fungal diseases and physiological disorders, which surely resulted from ripening. Many fungal diseases and physiological disorders appear on the fruit as soon as it becomes mature for consumption [4].

The lowest percentage of loss was observed for 'Idared', a cultivar considered as having good storability even in a normal atmosphere cold room [65], i.e., in conditions similar to those applied in this study. The highest percentage of losses caused by diseases and disorders (11.5%) and bruising (8.0%) was identified in 'Gala' apples after 14 days following vibration, which is quite understandable given the fact that 'Gala' also suffered the biggest drop in firmness in the same period (Table 3). 'Topaz' is sensitive to *Pezicula* sp. infection leading to bull's eye rot [66]—this was also confirmed by our study because 'Topaz' had the highest bull's eye rot incidence rate after 14 days of storage. 'Jonagold' and its mutants are characterized by excellent storability in a controlled atmosphere [67], but are far less storable in a normal atmosphere. The same will apply if apples suffer physical damage and are then stored at a relatively high temperature. The diagram shows that the treatment-induced losses in apples amounted to about 16% and were over 10% higher than the losses identified in fruit not subject to simulated transport vibration.

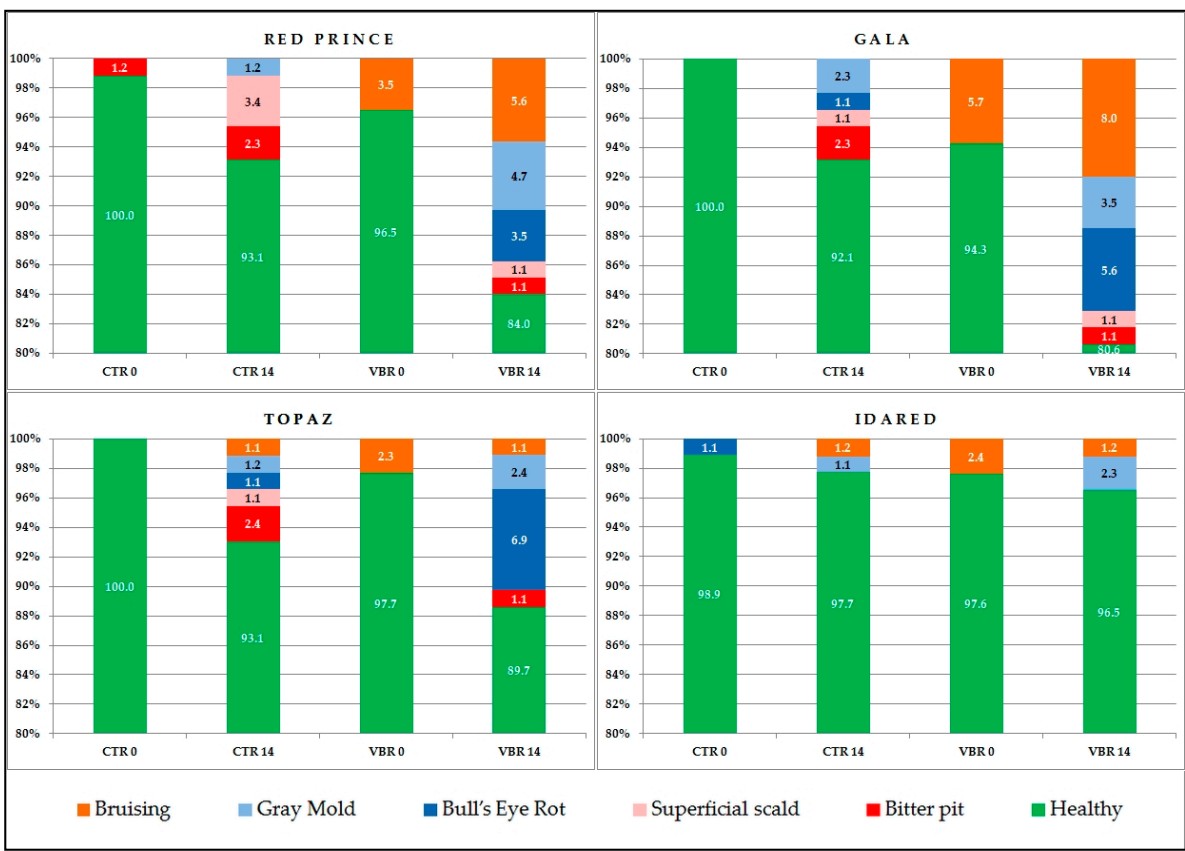

**Figure 3.** Influence of treatment on the incidence of physiological disorders, fungal diseases, and bruising. 1—Ctr 0, evaluation after harvest, day 0; 2—Vbr 0, evaluation after vibration, day 0; 3—Ctr 14, evaluation after 14-day storage; 4—Vbr 14, evaluation after vibration and 14-day storage.

## 4. Conclusions

In the study on four apple cultivars, designed under model transport conditions, the vibration frequency of 28 Hz resulted in the changes in some fruit parameters which occur also during storage. Those changes varied between the apple cultivars, but they often showed no clear trend or direction or were statistically insignificant. Skin colour varied depending on the treatment. However, there was no visible pattern of those differences, which was probably due to natural variations in raw material and a high share of red colouring (blush) in the total skin area. Flesh colour was relatively more stable across the samples, as reflected in small ΔE values within individual cultivars, compared to the high ΔE values obtained for the skin colour. Vibration caused a decrease in firmness, the biggest for 'Gala' apples and the smallest for 'Red Prince'. At the same time, a slight increase in TSS and dry matter was observed in a large part of the samples. However, those changes were mostly insignificant, especially in 'Red Prince' and 'Idared'. The pH value slightly decreased in most of the cultivars after either vibration or storage and showed a modest increase in the apples after vibration and storage. Changes in titratable acidity went in the direction opposite to those observed for pH. Mechanical vibration and storage led to an increase in the polyphenol content and antioxidant capacity of all studied cultivars. Within cultivars, the highest polyphenol content and antioxidant capacity were identified in the samples subjected to vibration followed by storage. No symptoms of fungal diseases or physiological disorders were found directly after applying mechanical vibration, but they were visible in both control and treated samples after 14-day storage, when the apples ripened. The lowest percentage of change was noted for 'Idared' and the highest for 'Gala'. The above results indicate that the values of the assessed parameters depended on the cultivar and varied differently, so it is not easy to indicate the best cultivar. However,

the parameter values suggest 'Red Prince' as the cultivar best retaining its quality during transport and storage (shelf-life) as it showed no significant changes in firmness, dry matter, and TSS and had a relatively high polyphenol content and antioxidant capacity. On the other hand, 'Topaz' was superior to 'Red Prince' in regard to total polyphenol content and antioxidant capacity, an 'Idared' had by far the lowest percentage of apples with diseases and physiological disorders after storage.

In sum, 'Red Prince' had the most stable basic quality parameters and a high polyphenol content and antioxidant capacity—this refers to the control and treated samples before and after storage, whereas 'Gala' ranked lowest along these criteria. Mechanical vibration at a frequency of 28 Hz and short-term storage did not reduce the high level of, and even increased, the antioxidant capacity of the analysed cultivars, which demonstrates that those apples remain healthy and valuable food and a good source of bioactive compounds even after transport and storage.

**Author Contributions:** Conceptualization, N.I. and D.W.-T.; methodology, D.W.-T. and N.I.; software, N.I. and K.B.; validation, G.P.Ł., and K.B.; formal analysis, D.W.-T.; investigation, D.W.-T. and N.I.; resources, G.P.Ł.; data curation, K.B.; writing—original draft preparation, D.W.-T. and N.I.; writing—review and editing, G.P.Ł.; visualization, K.B. and G.P.Ł.; supervision, G.P.Ł.; project administration, N.I.; funding acquisition, D.W.-T. and K.B. All authors have read and agreed to the published version of the manuscript.

**Funding:** This research received no external funding.

**Informed Consent Statement:** Not applicable.

**Data Availability Statement:** The data presented in this study are available on request from the corresponding author.

**Conflicts of Interest:** The authors declare no conflict of interest. The funders had no role in the design of the study; in the collection, analyses, or interpretation of data; in the writing of the manuscript, or in the decision to publish the results.

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
