# Peer review of "The Effect of Mechanical Vibration during Transport under Model Conditions on the Shelf-Life, Quality and Physico-Chemical Parameters of Four Apple Cultivars"

_agronomy, doi:10.3390/agronomy11010081_

Round 1

Reviewer 1 Report

What are the results of these studies? How will they help to transport the apples. The fact that they are damaged during transport has long been known.

It may be worth pointing out which of the cultivars are suitable for transport and so on.

 It is a pity that the experiment used old, tested cultivars that are of less and less economic importance - Idared, Topaz.The authors present the results in a very schematic way, indicating the results that the reader can read from the tables himself.  Lack of analysis of the results and solid discussion of the results.

The most ... was in variety x, the least in variety y. The aim of this work was to investigate changes in apples after storage and mechanical vibrations. This should be the main direction to discuss the results obtained.  Lack of analysis of these changes, reasons. Many characteristics were examined, but all these characteristics are discussed separately. No cause and effect relationship between the obtained results/measurements. Results should be discussed/comparable to controls.

The big disadvantage of this research is the very small number of objects examined: Three samples of 6 apples each were randomly selected from the experimental batch of each cultivar for the analysis of physical and chemical parameters. The small number of objects may have influenced some of the results - how to explain that after a period of simulated transport the amount of e.g. polyphenols (Idared, Topaz) increased significantly. Please indicate how this biochemical process influenced it.

You should not conclude on the basis of 18 fruits, which were only tested in one year

17-22:  Quality changes in apples were evaluated based on skin and flesh colour, total soluble solids, dry matter, firmness, titratable acidity, pH value, total polyphenol content and antioxidant capacity. The applied vibrations at a frequency of 28 Hz caused changes in the above parameters, which were visible also after storage and depended on cultivar, but did not show any clear trend or direction. Skin colour varied whereas flesh colour remained stable.

Not precisely - it changed the above parameters, but the colour was stable?

26: polyphenol content and antioxidant capacity increased...

It has increased, but why? Perhaps there has been a concentration of compounds, which may be indicated by an increase in dry matter content.

90: Was the fruit close/similar in size? This affects many parameters, including polyphenols. Smaller fruit is more rich in these compounds.

200: 'The results were analysed by one-way and two-way ANOVA...' - which results were analysed in what way. No information in tables.

266: the same letters in lines or columns?

No units in Tables 4 and 5.

437: Conclusions require a fundamental change. There is inconsistent information.  No short, general summary.

452: Miscellaneous information in Abstract '...changes in the above parameters...' and Conclusions 'Mechanical vibrations at a frequency of 28Hz and short-term storage did not reduce the antioxidant activity and thus the health promoting properties of the analysed cultivars... Please explain.

Author Response

Answer for reviewer 1

Remark

Answer

What are the results of these studies? How will they help to transport the apples. The fact that they are damaged during transport has long been known.

It is known that mechanical vibrations during transport damage fruit and other raw materials, but there are no studies on the influence of transport of apples on their antioxidant properties.

Despite changes in firmness or concentration of the basic nutrients (refractometric extract components), there was no decrease in the content of antioxidant compounds in the vibrated apples, on the contrary, there was an increase of compounds.

This makes it possible to conclude that apples retain their high antioxidant potential after transport, and therefore also health-promoting, despite the deterioration of other physico-chemical parameters.  Thanks to this, as fruits with good storage properties, apples are a good source of antioxidant compounds in the off-season, winter and spring.  These explanations are added in abstract, conclusion and chapter 3.4.

It may be worth pointing out which of the cultivars are suitable for transport and so on.

All changes or fluctuations in the parameters tested depended on the cultivar. Therefore, the indication of the best cultivar, least sensitive to changes after transport or storage, should be considered in terms of individual parameters, as described in the discussion of the results and in the conclusions.  For example, the cultivar 'Red Prince' showed the best stability of firmness (row 286), relatively high polyphenol content and antioxidant activity (row 529), but was characterizing, in addition to the 'Gala' variety, a high share of fruit with disease symptoms (row 494-495).  The greatest resistance to damage and disease was demonstrated by the cultivar 'Idared'. However, taking all the characteristics into account, 'Red Prince' was identified as the most stable cultivar and the worst was 'Gala' (row 534-536).

 It is a pity that the experiment used old, tested cultivars that are of less and less economic importance - Idared, Topaz.

All four varieties are of great economic importance in many countries. Remark to the age of varieties is controversial.  Golden Delicious, for example, was bred in 1890. Is it not important for this reason, since it is one of the most popular in the world?  Gala and Jonagold mutants are currently the most popular in the world.  The Topaz variety was obtained almost a hundred years later than Golden (in 1984) and is the most often planted cultivar in Europe in organic orchards.

The authors present the results in a very schematic way, indicating the results that the reader can read from the tables himself.  Lack of analysis of the results and solid discussion of the results.

The discussion of the results removed the numerical values that can be read in the tables, while the discussion of the results in the different parts of Chapter 4 was complemented by additional literature.

The most ... was in variety x, the least in variety y. The aim of this work was to investigate changes in apples after storage and mechanical vibrations. This should be the main direction to discuss the results obtained.  Lack of analysis of these changes, reasons. Many characteristics were examined, but all these characteristics are discussed separately. No cause and effect relationship between the obtained results/measurements. Results should be discussed/comparable to controls.

The discussion of the results was complemented by the effect of mechanical storage and vibration on the individual parameters tested.  After discussing the results, a text was added summarizing the results for the control sample and describing the correlation between the parameters (e.g. row 438-452).  In chapters Results and Discussion, additional phrases have been introduced explaining the changes in the test parameters related to maturation, storage and transport, based on additional literature.

The big disadvantage of this research is the very small number of objects examined: Three samples of 6 apples each were randomly selected from the experimental batch of each cultivar for the analysis of physical and chemical parameters. The small number of objects may have influenced some of the results - how to explain that after a period of simulated transport the amount of e.g. polyphenols (Idared, Topaz) increased significantly. Please indicate how this biochemical process influenced it.

According to the authors, the evaluation of the parameters tested on a sample of 3 groups of 6 apples (18 fruits together for treatment) is sufficient.  For example, 'Red Prince' apples with an average weight of 230g in the amount of 18 fruit are about 4kg, which is 26% of the weight of the fruit of each treatment.

The increase in the polyphenol content after wibration, in the range of 1-8% relative to the control sample, is most likely the result of the physiological response of the plant raw material to stress conditions during simulated transport, as is the case as a result of fruit damage e.g. during harvesting.  The discussion took into account possible mechanisms for increasing the polyphenol content of the apples tested (e.g. row 431-437).

You should not conclude on the basis of 18 fruits, which were only tested in one year

Given that the seasonal variability of the raw material is very important, research will continue in the following years, also using other cultivars and species of fruit, as there is literature on the subject.

17-22:  Quality changes in apples were evaluated based on skin and flesh colour, total soluble solids, dry matter, firmness, titratable acidity, pH value, total polyphenol content and antioxidant capacity. The applied vibrations at a frequency of 28 Hz caused changes in the above parameters, which were visible also after storage and depended on cultivar, but did not show any clear trend or direction. Skin colour varied whereas flesh colour remained stable.

In fact, there is a lack of consistency in those findings, as is apparent from the generalisation used in Abstact.  This was clarified as the changes were found in most of the parameters tested, but the changes were often statistically insignificant or did not show clear targeted trends (row 20-21).

Not precisely - it changed the above parameters, but the colour was stable?

In abstract and discussion of the results, this is clarified (row 22-23, chapter. 3.1, 254-255).  On the one hand, changes were observed, sometimes statistically significant, in the colour parameter values of the cultivar, after shaking or storage, but on the other hand, these changes were not found to show a uniform direction (increase or decrease in value), and were therefore linked rather to the output variation of the colour of the raw material.  The fruit used in the research was standardised in accordance with the international OECD standard (row 99). Accordingly, the apples tested belong to groups B and C, in which blush accounts for at least 1/2 and 3/4 of the peel surface, respectively. With such extensive variable blush, the measurement of the basic color of the peel is not repeatable, hence the large variation in the value of the color parameters.  on the individual fruits (line 224-227).  On the other hand, the colour of the flesh was relatively even, which was found on the basis of the low values of the coefficient ΔE (0,3-2), which shows that for the average observer these changes were invisible (row 20-22; 250-256).

26: polyphenol content and antioxidant capacity increased...

The increase in polyphenol content was most statistically significant, in the range of 1-8% after shaking and 5-14% after storage, relative to the control. It was found in all cultivars and treatments, which is rather not a matter of chance, but most likely a phenomenon associated with biochemical transformations in fresh raw material, in which physiological processes and reactions to external stimuli (such as mechanical vibrations) take place.  The metabolic activity of apples, as climacteric fruits, also accelerates the production of ethylene (row 383), especially under stressful conditions.

It has increased, but why? Perhaps there has been a concentration of compounds, which may be indicated by an increase in dry matter content.

Fluctuations in the dry matter content depending on the treatment  within the apple cultivar were in any case statistically insignificant, as was the case for most samples in terms of TSS content (Table 4).  Thus, the increase in polyphenol content is not due to an increase in the dry matter content and density of the ingredients.  The polyphenol content is shown per p.m. to make the differences between the test cultivars more visible, but the increase in polyphenol content was also recorded in fresh raw material weight (results not published). The discussion complemented various other possible mechanisms/reasons for the increase in polyphenol content during maturation, storage or transport of apples (row 431-437).

90: Was the fruit close/similar in size? This affects many parameters, including polyphenols. Smaller fruit is more rich in these compounds.

The studies used extra-grade fruit in accordance with OECD standards (row 99), which means that the diameter of the apples of the cultivar could not differ by more than 5 mm.  Thus, the effect of fruit size on the polyphenol content was rather small.

200: 'The results were analysed by one-way and two-way ANOVA...' - which results were analysed in what way. No information in tables.

The results of parameters such as d.m. content, TSS, acidity, pH values and firmness were analysed using a single-factor variance analysis and the polyphenol content and antioxidant activity using a two-way factor.  This information is supplemented by footnotes below the tables.

266: the same letters in lines or columns?

The significance of statistical differences in all analyses was considered within a given cultivar, which is included in the footnotes under each table  (average values marked with the same letter are not statistically different at p = 0.05 within one cultivar).  In row 287, the differences apply to variants for a given cultivar in rows, as in Table 3.

No units in Tables 4 and 5.

All measuring units are given in the methodology. According to the Reviewer's note, units have been added to Table 4, 5 and 6.

437: Conclusions require a fundamental change. There is inconsistent information.  No short, general summary.

The conclusions have been amended and revised to include a brief summary, generalisation of the results obtained.

452: Miscellaneous information in Abstract '...changes in the above parameters...' and Conclusions 'Mechanical vibrations at a frequency of 28Hz and short-term storage did not reduce the antioxidant activity and thus the health promoting properties of the analysed cultivars... Please explain.

Abstract and conclusion have been thoroughly corrected so that there is no conflicting information.  In the conclusions, we wanted to point out that, despite the deterioration of certain parameters, not only did the baseline level of polyphenols and antioxidant activity remain in all apples, but even increased them, which makes it possible to conclude that these fruits can be a good source of polyphenols, not only immediately after harvesting, but also after storage. This is of great importance in countries where the fresh fruit season is relatively short.

The authors thank you for your thorough review. We have included all the comments in the text or explained them in the above reply.

Reviewer 2 Report

The manuscript titled: ‘The effect of mechanical vibrations during transport 2 under model conditions on the shelf-life, quality and 3 physico-chemical parameters of four apple cultivars’ was reviewed. The manuscript discussed effect of vibration on fruit quality and shelf file of four apple cultivars.

There are quite a large number of grammatical errors in the manuscript. The articles (a, the) are not used or not used properly. Therefore, it needs to be edited for English. Some of the errors are mentioned but not necessarily all of them.

Tables should come after text, but they all came before text.

References can be reduced and only newer ones to be kept unless really necessary.

In M&M, there is no discussion as if diseases were identified, but in results they are.

Based on the concerns mentioned here and in the manuscript, it needs major revision.

Author Response

Answer for reviewer 2

Remark

Answer

There are quite a large number of grammatical errors in the manuscript. The articles (a, the) are not used or not used properly. Therefore, it needs to be edited for English. Some of the errors are mentioned but not necessarily all of them.

The tables are moved to the end of the text with an overview of each result.

Tables should come after text, but they all came before text.

The tables are moved to the end of the text with an overview of each result.

References can be reduced and only newer ones to be kept unless really necessary.

According to the authors, the number of references is appropriate and necessary to discuss the results, while in the "Instruction for authors" there are no restrictions in this regard.  In recent articles in Agronomy, the number of references ranges from 40 to more than 80.

In M&M, there is no discussion as if diseases were identified, but in results they are.

It is difficult to understand the reviewer's remarks with certainty, but I note that in M&M point 2.4.9 there was a description of assessing the prevalence of disorders and diseases.

Based on the concerns mentioned here and in the manuscript, it needs major revision.

The work introduced amendments and additions to the analysis and discussion of the results and conclusions, and corrected the layout of the tables in relation to the text, as guidelines of the Reviewer.  All changes to the manuscript are marked in red.
